# SCENEFORGE: Enhancing 3D-text alignment with Structured Scene Compositions

**Cristian Sbrolli**
Department of
Electronics, Information and Bioengineering
Politecnico di Milano
Via Ponzio 34/5, 20133 Milan, Italy
cristian.sbrolli@polimi.it

**Matteo Matteucci**
Department of
Electronics, Information and Bioengineering
Politecnico di Milano
Via Ponzio 34/5, 20133 Milan, Italy
matteo.matteucci@polimi.it

## Abstract

The whole is greater than the sum of its parts, even in 3D-text contrastive learning. We introduce SCENEFORGE, a novel framework that enhances contrastive alignment between 3D point clouds and text through structured multi-object scene compositions. SCENEFORGE leverages individual 3D shapes to construct multi-object scenes with explicit spatial relations, pairing them with coherent multi-object descriptions refined by a large language model. By augmenting contrastive training with these structured, compositional samples, SCENEFORGE effectively addresses the scarcity of large-scale 3D-text datasets, significantly enriching data complexity and diversity. We systematically investigate critical design elements, such as the optimal number of objects per scene, the proportion of compositional samples in training batches, and scene construction strategies. Extensive experiments demonstrate that SCENEFORGE delivers substantial performance gains across multiple tasks, including zero-shot classification on ModelNet, ScanObjNN, Objaverse-LVIS, and ScanNet, as well as few-shot part segmentation on ShapeNetPart. SCENEFORGE's compositional augmentations are model-agnostic, consistently improving performance across multiple encoder architectures. Moreover, SCENEFORGE improves 3D visual question answering on ScanQA, generalizes robustly to retrieval scenarios with increasing scene complexity, and showcases spatial reasoning capabilities by adapting spatial configurations to align precisely with textual instructions.

## 1 Introduction

Large-scale contrastive learning has transformed vision-language modeling, with early breakthroughs like CLIP [19] and ALIGN [11] demonstrating the power of aligning visual and textual representations at scale. By leveraging vast image-text datasets, these models have achieved remarkable success in zero-shot recognition, retrieval, segmentation, and transfer learning. Following these advancements in 2D, researchers have increasingly turned to 3D, where richer geometric and spatial information is critical for robotics, virtual environments, and augmented reality. However, scaling contrastive learning to 3D remains challenging due to the limited availability of large-scale datasets. Recent works such as Uni3D [31] and OmniBind [21] have made significant strides by leveraging OpenShape [17] dataset, a large-scale ensemble of 3D-text data. These methods align 3D point clouds with 2D-text representation spaces using pretrained CLIP models, achieving strong zero-shot performance on single-object classification benchmarks like ModelNet and ScanObjNN. However, despite these advances, the available 3D text data remain limited, specially compared to image-text datasets, necessitating new strategies to enhance learning.

39th Conference on Neural Information Processing Systems (NeurIPS 2025).

In this work, we propose a novel approach which allows us to both virtually increase the amount of 3D-text training data and introduce harder samples, improving the contrastive representation. Inspired by image compositions typically used for augmenting classification datasets and methods in previous works, we leverage compositional learning in 3D by constructing multi-object multimodal training samples for contrastive learning. Our method is motivated by two key insights. First, unlike 2D images, where objects are inherently tied to backgrounds, lighting conditions, and perspectives, individual 3D point clouds can be freely combined into structured scenes without visual artifacts. Second, the spatial flexibility of 3D data allows explicit control over object positioning, an ability difficult to achieve in 2D. In contrast to images, 3D objects exist independently of any scene context, enabling meaningful spatial configurations and natural textual descriptions with relational cues (e.g., "A on top of B"). We further refine these descriptions using a large language model, generating diverse and nuanced combinations. Using structured synthesis, we virtually leverage a large-scale, synthetic multi-object 3D-text dataset grounded in real-world captions, significantly expanding diversity and complexity. Training any 3D encoder on these compositional scenes to align with CLIP's representation space achieves consistent improvements across tasks including zero-shot classification, retrieval, segmentation, and VQA. We empirically demonstrate the model-agnostic nature of our compositional augmentations, confirming their effectiveness across multiple encoder architectures. Our contributions are threefold: (1) proposing a novel compositional data pipeline for 3D-text contrastive learning approach synthesizing multi-object 3D scenes; (2) demonstrating consistent performance gains across tasks and backbones; and (3) analyzing the impact of key design choices, including composited object counts, object ratios, and 3D composition strategies.

## 2 Related Work

Among the first successful works on contrastive alignment of 3D data, ULIP [23] aligned 3D features with CLIP and scaled to the Objaverse dataset in ULIP-2 [23], while Uni3D [31] further advanced this direction by scaling to billion-parameter models, leveraging 2D pretraining and using OpenShape [17] ensembled data. Other works proposed methods for improving the alignment: TAMM [30] mitigates the domain gap between rendered and natural images via adapter modules, while MixCon3D [8] sculpts holistic 3D representations by integrating multi-view rendered images and point clouds. OmniBind [21] adpots instead a differente approach, proposing to ensemble multiple pretrained models via a learnable routing mechanism, achieving state-of-the-art multimodal performance. Our work instead explores an orthogonal approach, scaling multimodal learning by virtually increasing dataset diversity through composition-based augmentation. This strategy aims to improve generalization by exposing the model to a richer distribution of multimodal data.

**Compositionality in Multimodal Learning.** Compositionality refers to the idea of constructing new concepts by combining simpler ones, a principle that is fundamental human-like concept generalization [12]. Multi-sample composition augmentations such as CutMix [26] and MixUp [28] have been demonstrated to be highly effective for robustness and generalization in 2D tasks such as classification, detection, and segmentation. In the 3D domain, adaptations of these augmentation strategies have been proposed and shown to be effective for unimodal 3D tasks, such as point cloud classification and segmentation [3, 29]. More recently, a stronger form of composition has been employed in image-text contrastive learning [1], demonstrating improved multimodal alignment. Unlike CutMix and MixUp, which blend two images by either mixing pixel values or pasting cut regions, thus being only a weak form of compositional learning, this approach vertically stacks centered crops of two images and combines their captions using the conjunction "and", creating stronger semantic compositions. Building on these ideas, we extend compositional augmentation to 3D data, where the structural properties of point clouds make them particularly well-suited for such techniques. Unlike images, which often contain noisy elements such as background or secondary objects, point clouds representing individual objects can be seamlessly merged into a unified scene without introducing visual artifacts. Moreover, 3D compositional augmentations enable explicit modeling of spatial relationships, which can be reflected in textual descriptions, facilitating relational reasoning alongside object-level recognition.

## 3 Method

We introduce SCENEFORGE (SF), a framework for composing multi-object 3D scenes by combining individual point clouds with their text descriptions according to explicit spatial relations. SCENE-

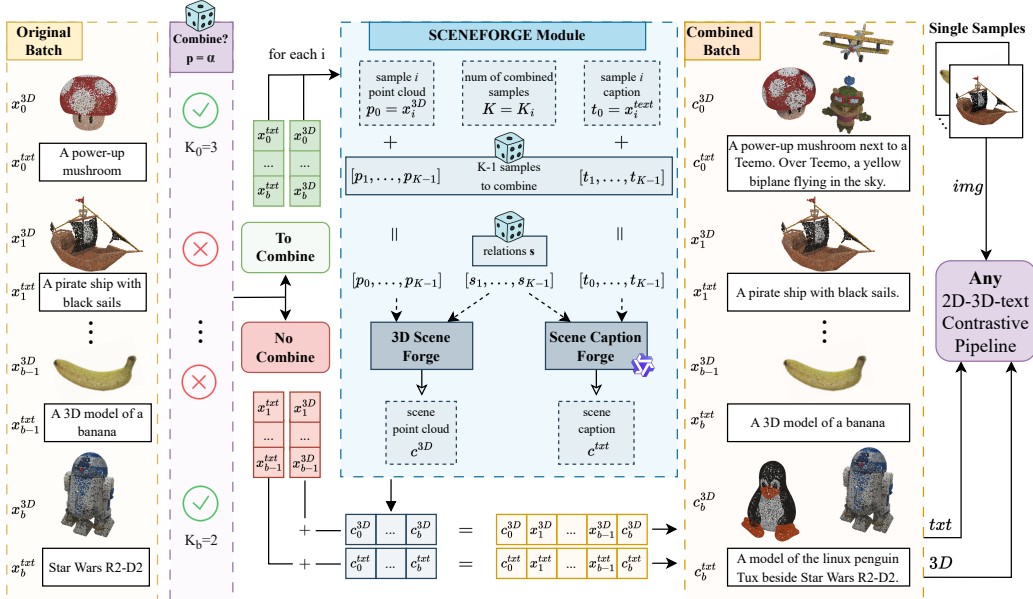

**Figure 1: Our multimodal scene composition framework for contrastive 3D-text learning.**
Given a batch of point clouds and their captions, each sample is randomly kept as single object or combined into a synthetic 3D scene with a random number of objects. Objects designated for combination are passed to the SCENEFORGE module, which samples the additional objects and their captions. Spatial relationships among the selected objects are randomly assigned, and the *3D Scene Forge* and *Scene Caption Forge* generate the corresponding combined 3D scene and its composite caption. The newly formed scenes are merged with the unmodified objects to construct a final batch, which is then used for contrastive 3D-text alignment using a pretrained CLIP text encoder.

FORGE can be incorporated into any contrastive multimodal 3D–text learning pipeline, and we refer to the resulting approaches as SF-variants. An overview of our pipeline is shown in Figure 1.

## 3.1 SCENEFORGE module

SF takes as input a (point cloud, caption) sample $(p_0, t_0)$ and the number $K$ of objects to be combined. It randomly samples the required $K - 1$ point clouds $\{p_1, p_2, \ldots, p_{K-1}\}$ and their corresponding textual captions $\{t_1, t_2, \ldots, t_{K-1}\}$ to generate a composite 3D scene $c^{3D}$ and an associated scene caption $c^{txt}$. The framework consists of two core modules: the *3D Scene Forge*, which spatially arranges the point clouds, and the *Scene Caption Forge*, which constructs the textual description of the generated scene. Both modules rely on a set of randomly sampled spatial relations $\{s_1, s_2, \ldots, s_{K-1}\}$, which define the relative placement of each object with respect to the one previously placed. These relations dictate how each object is positioned within the combined 3D scene and are used to generate a caption that accurately describes the spatial arrangement. The spatial relations are sampled once per scene and remain consistent across both the 3D and captioning processes.

**Spatial Relations.** We define three simple spatial relations: *"over," "under,"* and *"next to"*. Since objects in the OpenShape dataset are not consistently oriented along the horizontal axes, using directional terms such as "left" or "right" would be ambiguous unless interpreted as absolute displacements along the X or Z axis. However, this would limit compositions to strictly axis-aligned translations. Instead, we employ the *"next to"* relation, which allows flexible horizontal placement while preserving semantic coherence. In contrast, the vertical orientation of objects is consistent, making *"over"* and *"under"* well-defined, provided that appropriate constraints are imposed on rotation augmentations, as discussed later (Section 3.2). These relations serve as constraints for both the spatial placement of objects in the 3D scene and the construction of descriptive captions that reflect the compositions.

**3D Scene Forge.** The 3D Scene Forge arranges objects according to their assigned spatial relations, ensuring a semantically coherent composition. The high-level procedure is outlined in Algorithm 1. The module takes as input $K$ point clouds, a set of $K - 1$ pairwise spatial relations, and a target size

**Input:** Samples $p$, Relations $s$, Target count $P$
**Output:** Composed 3D sample $c^{3D}$
$c^{3D}, p_{prev} \leftarrow \mathcal{A}^{3D}(p_0)$
**for** $i = 1$ *to* $n$ **do**
 $\quad p_i \leftarrow \mathcal{A}^{3D}(p_i)$
 $\quad \Delta_{pos} \leftarrow \mathcal{P}(p_i, p_{prev}, s_i)$
 $\quad p_i \leftarrow p_i + \Delta_{pos} + \delta + \epsilon$
 $\quad c^{3D} \leftarrow \mathrm{cat}(c^{3D}, p_i)$
 $\quad p_{prev} \leftarrow p_i$
**end**
$c^{3D} \leftarrow \mathcal{A}^{3D}(\mathrm{subsample}(c^{3D}, P))$
**return** $c^{3D}$

**Algorithm 1**: 3D Scene Forge algorithm.

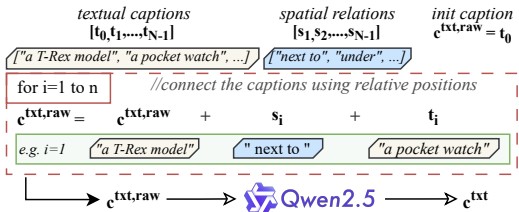

Figure 2: **Scene Caption Forge.** Starting from the initial caption ($t_0$), each caption ($t_i$) is connected using its relative position ($s_i$), creating a raw combined caption $c^{\text{txt,raw}}$. The raw caption $c^{\text{txt,raw}}$ is then refined to the final $c^{\text{txt}}$ using Qwen2.5.

for the final merged cloud. The first object $p_0$ is used to initialize the scene $c^{3D}$. Each subsequent object $p_i$ is then placed relative to the previous object $p_{i-1}$ according to its assigned spatial relation $s_i$, using a function $\mathcal{P}(\cdot)$ that computes the appropriate displacement. Specifically, we define $\mathcal{P}(\cdot)$ based on the spatial relation between objects. For **"over"**, we align the minimum $z$-coordinate of $p_i$ above the maximum $z$-coordinate of $p_{i-1}$:

$$\mathcal{P}(p_i, \, p_{i-1}, \, \text{"over"}) = \max_{\mathbf{z}}(p_{i-1}) - \min_{\mathbf{z}}(p_i),$$

and reverse the roles for **"under"**. For **"next to"**, we sample a horizontal unit vector $\mathbf{d}$ in the $xy$-plane and compute:

$$\mathcal{P}(p_i, \, p_{i-1}, \, \text{"next to"}) = \left( \max_{\mathbf{x} \in p_{i-1}} \langle \mathbf{x}, \mathbf{d} \rangle - \min_{\mathbf{y} \in p_i} \langle \mathbf{y}, \mathbf{d} \rangle \right) \mathbf{d},$$

where $\langle \cdot, \cdot \rangle$ is the inner product.

In all relations, to prevent perfect alignment and introduce slight randomness, we add a fixed offset $\delta$ along the shift direction ($+\mathbf{z}$, $-\mathbf{z}$ or $\mathbf{d}$), along with a small Gaussian noise term $\epsilon \in \mathcal{R}^3$. After placing all $K$ objects, we downsample the composite point cloud $c^{3D}$ to the target number of points $P$, ensuring diversity while maintaining spatial consistency. Notice that normalization and augmentation $\mathcal{A}^{3D}$ is performed on each sample before adding it to the scene, as well as on the final scene.

**Scene Caption Forge** This module constructs a textual description of the combined scene by sequentially incorporating individual object captions $\{t_0, t_1, \ldots, t_{K-1}\}$ and their corresponding spatial relations $\{s_1, s_2, \ldots, s_{K-1}\}$. The captioning process mirrors the spatial composition of the 3D scene, starting with the first object's caption and appending each subsequent caption preceded by its respective spatial relation. However, due to the method's simplicity, the generated caption may exhibit artifacts such as misplaced punctuation before spatial relations, incorrect capitalization following conjunctions, and disfluent sentence structures. Consequently, we refer to this as a raw caption, $c^{txt,raw}$. To enhance readability and coherence, we refine the raw caption using a large language model (Qwen2.5 [24]), obtaining the refined caption $c^{txt}$. The model corrects grammar, punctuation, and structure while preserving the original meaning and spatial relationships. It restructures the text into a fluent, human-like description, splitting overly long sentences when necessary. Beyond improving fluency, this rewriting process also enhances caption diversity and refines OpenShape [17] captions, originally generated by BLIP [14] and Microsoft Azure Cognitive Services (2023), by leveraging the more advanced linguistic capabilities of a recent language model. The full prompt used for refinement is provided in the supplementary, together with an ablaton on the LLM. An overview of the proposed method is provided in Figure 2.

### 3.2 Training Scheme

We mix single and multi-object samples in the same training batch with a predefined ratio $\alpha$ (ablated in Section 5). This allows the model to retain strong performance on single-object tasks while benefiting from the additional compositional training signals. For each combined sample, the number of objects to combine is randomized between 2 and the maximum number of combinable objects $N$, which we investigate in Section 4. This allows for $\sum_{k=1}^{N} \frac{D!}{(D-k)!} 3^{k-1}$ possible configurations with

up to $k$ samples and 3 possible relations, with $D$ being the dataset cardinality (extended derivation in the supplementary). With OpenShape data, this allows for $6E$ scenes.

Our framework operates entirely at the batch–generation level and can be plugged into *any* contrastive pipeline that aligns text, image, and 3D point-cloud embeddings. Because all previous methods include images, we likewise keep the 2D modality when deploying SCENEFORGE. In principle, we could rasterise every composed scene and align its image views, but real-time rendering is computationally prohibitive given our budget. Instead, we mask composed samples for the 2D–3D loss terms, evaluating those terms only on pre-rendered single-object views. This preserves baseline performance on image–3D tasks (see supplementary) while allowing our compositions to focus on strengthening text–3D alignment. Following established practice, the CLIP image and text encoders stay frozen; gradients flow only through the 3D encoder.

**Loss Partitioning.** We consider contrastive models employing the InfoNCE loss proposed in CLIP [19]. For modalities $m, n \in \{txt, 2\mathrm{D}, 3\mathrm{D}\}$ and a sample subset $\mathcal{S}$, we define

$$\mathcal{L}_{m \to n}(\mathcal{S}) = -\frac{1}{|\mathcal{S}|} \sum_{i \in \mathcal{S}} \log \frac{\exp(\langle e_i^m, e_i^n \rangle / \tau)}{\sum_{j \in \mathcal{S}} \exp(\langle e_i^m, e_j^n \rangle / \tau)}, \tag{1}$$

where $e_i^m, e_i^n$ are $\ell_2$-normalised embeddings and $\tau$ is a learnable temperature.

Let $\mathcal{S}_c$ and $\mathcal{S}_s$ denote the composed and single-object samples in a batch, with $N = |\mathcal{S}_c| + |\mathcal{S}_s|$. Because each sample is composed with probability $\alpha$, $\mathbb{E}[|\mathcal{S}_s|] = (1-\alpha)N$. We scale the image–3D block so that, *per batch*, it contributes the same total gradient budget as the text–3D block:

$$\mathcal{L} = \underbrace{\tfrac{1}{2}\big[\mathcal{L}_{3\mathrm{D} \to txt}(\mathcal{S}_c \cup \mathcal{S}_s) + \mathcal{L}_{txt \to 3\mathrm{D}}(\mathcal{S}_c \cup \mathcal{S}_s)\big]}_{\text{text–3D (all } N \text{ samples)}} + \frac{N}{|\mathcal{S}_s|} \underbrace{\tfrac{1}{2}\big[\mathcal{L}_{3\mathrm{D} \to 2\mathrm{D}}(\mathcal{S}_s) + \mathcal{L}_{2\mathrm{D} \to 3\mathrm{D}}(\mathcal{S}_s)\big]}_{\text{2D–3D (singles only)}}. \tag{2}$$

Considering the large batch sizes used in contrastive models, $|\mathcal{S}_s|$ is tightly concentrated around its mean, so we simply replace the dynamic factor $N/|\mathcal{S}_s|$ by its expectation $1/(1-\alpha)$.

**Implementation Details.** We instantiate our scene–composition pipeline on three approaches using different point-cloud encoders: OpenShape-PointBERT [17], Uni3D-G [31], and ViT-Lens-G (a frozen ViT-bigG/14 with trainable adapter lenses) [13]. This variety allows us to verify the encoder-agnostic nature of our method and to select the approach which benefit the most from our pipeline. All variants are trained with their public available code and our modified loss for 200 epochs with a global batch size of 1152, $\alpha = 0.5$, and a target point-cloud resolution of $P = 10\,\mathrm{k}$ points. This point budget was chosen as a trade-off between detail and efficiency, as detailed in the supplementary. For caption generation, we use the *Qwen 2.5 7B-instruct* [24] large language model. SCENEFORGE only requires an additional GPU hosting the lightweight LLM for composition. We pre-generate the first $M$ batches and then switch to a producer–consumer setup: while batch $t$ trains, batch $t+M$ is assembled in parallel. Though composition latency is not always fully hidden, parallelization significantly reduce it. For faster models, multiple SCENEFORGE instances and quantization can be used to further amortize overhead. Measured slowdown ranges from 0% to 50% depending on backbone, LLM quantization and replication (see supplementary). For the function $\mathcal{A}^{3D}$, we normalize point clouds to the unit sphere and apply random point dropout and scaling. Rotation and translation strategies differ based on whether we process single objects, objects intended for combination in a scene, or the final composed scene. For single objects, we allow both shifts and larger rotations. However, for objects that will be combined, shifts are disabled to avoid inconsistencies in composition, while full rotations around the vertical axis and slight rotations along other axes are permitted. The latter ensures that concepts such as "over" and "under" remain semantically meaningful. For the final combined point cloud, we adopt the same rotation constraints as in the previous case to preserve spatial semantics but additionally allow translations.

## 4 Experiments

### 4.1 Zero-Shot Classification

**Datasets.** We follow the standard zero-shot evaluation protocol on Objaverse LVIS [6], Model-Net40 [22] and ScanObjNN [20], where categories are mapped to text prompts by formatting a set of

| Model | LVIS | | ModelNet | | ScanObjNN | | Scannet | Avg Δ |
|---|---|---|---|---|---|---|---|---|
| | T1 | T5 | T1 | T5 | T1 | T5 | T1 | |
| ULIP 2 | 46.3 | 75.0 | 84.0 | 97.2 | 45.6 | 82.9 | 38.1 | – |
| TAMM | 42.0 | 71.7 | 86.3 | 98.1 | 56.7 | 86.1 | 42.4 | – |
| MixCon3D | 47.5 | 76.2 | 87.3 | 98.1 | 57.7 | 89.8 | 43.0 | – |
| OmniBind-L | – | – | – | – | – | – | – | – |
| OmniBind-F | – | – | – | – | – | – | – | – |
| OpenShape | 39.1 | 68.9 | 85.3 | 97.4 | 47.2 | 84.7 | 40.3 | +1.50 |
| SF-OpenShape | 41.7 | 71.5 | 86.7 | 98.1 | 48.0 | 85.9 | 41.5 | |
| ViT-Lens | 50.1 | 78.1 | 86.8 | 97.8 | 59.8 | 87.7 | 43.8 | +0.78 |
| SF-ViT-Lens | 50.9 | 78.4 | 87.3 | 98.0 | 60.9 | 89.1 | 44.5 | |
| Uni3D | 47.2 | 76.1 | 86.8 | 98.4 | 66.5 | 90.1 | 43.9 | +1.73 |
| SF-Uni3D | 48.9 | 78.4 | 87.5 | 99.0 | 67.3 | 91.5 | 47.6 | |

(a) Trained on ensemble (no LVIS).

| Model | LVIS | | ModelNet | | ScanObjNN | | Scannet | Avg Δ |
|---|---|---|---|---|---|---|---|---|
| | T1 | T5 | T1 | T5 | T1 | T5 | T1 | |
| ULIP 2 | 50.6 | 79.1 | 84.7 | 97.1 | 51.5 | 89.3 | 38.9 | – |
| TAMM | 50.7 | 80.6 | 85.0 | 98.1 | 55.7 | 88.9 | 41.8 | – |
| MixCon3D | 52.5 | 81.2 | 86.8 | 98.3 | 58.6 | 89.2 | 44.1 | – |
| OmniBind-L | 54.0 | 82.9 | 86.6 | 99.0 | 64.7 | 94.2 | 46.3 | – |
| OmniBind-F | 53.6 | 81.8 | 87.1 | 99.0 | 64.7 | 94.4 | 46.1 | – |
| OpenShape | 46.8 | 77.0 | 84.4 | 98.0 | 52.2 | 88.7 | 39.4 | +1.43 |
| SF-OpenShape | 48.1 | 78.4 | 85.2 | 98.3 | 53.4 | 89.5 | 41.8 | |
| ViT-Lens | 52.0 | 79.9 | 87.6 | 98.4 | 60.1 | 90.3 | 43.7 | +0.85 |
| SF-ViT-Lens | 52.8 | 80.7 | 88.0 | 89.9 | 60.9 | 91.2 | 45.1 | |
| Uni3D | 53.5 | 82.0 | 87.3 | 99.2 | 63.9 | 91.7 | 45.8 | +1.75 |
| SF-Uni3D | 54.7 | 84.8 | 88.2 | 99.2 | 65.2 | 93.4 | 49.4 | |

(b) Trained on ensemble (with LVIS).

Table 1: **Zero-shot classification accuracy (%).** "SF-" denotes models trained with SCENEFORGE. Green cells ( ) are the best results, yellow ( ) the second best. The rightmost column reports the average Top-1 improvement (Δ) of the augmented model over its baseline.

templates (e.g., "a point cloud model of a ) and the model is evaluated on the classification accuracy. Additonally, adopting the pipeline from CLIP[2] [27], we test our models on the Scannet [5] dataset to evaluate their zero-shot performance on object instances from real-world scenarios.

**What is the optimal value of N?** Figure 3 tracks zero-shot accuracy for three contrastive learners: OpenShape-PointBERT [17], Uni3D (EVA-Giant) [7, 31], and ViT-Lens-G [13], as we vary the maximum number of shapes that SCENEFORGE can merge during training. A consistent trend emerges. From $N=1$ to $N=3$, accuracy rises monotonically: LVIS gains $+0.8 - 1.3$ pp, ScanObjNN $+0.8 - 1.3$ pp, Scan-Net $+1.4 - 3.6$ pp, and ModelNet40 about $+0.8$ pp. Improvements peak at $N=3$; Uni3D benefits most, while the lighter OpenShape also advances. ViT-Lens shows a more modest gain, plausibly because its frozen CLIP backbone trains only lightweight adapters, offering less plasticity when confronted with composed shapes. Increasing to $N=4$ plateaus on canonical datasets and already trims accuracy on LVIS and ScanNet, whose higher intra-class variability makes them clutter-sensitive. At $N=5$ the drop is universal, indicating that squeezing five objects into a fixed 10 k-point budget fragments salient geometry and introduces caption noise,

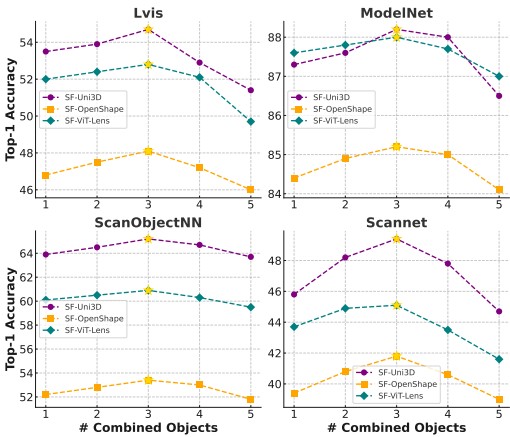

Figure 3: Top-1 accuracy across different datasets (Lvis, ModelNet, ScanObjectNN, and Scannet) as a function of the number of combined objects.

hampering alignment. Interestingly, performance variations are relative minor on ModelNet40 and ScanObjNN, but the effect is far more pronounced on LVIS and ScanNet. We attribute this to the latter's more complex object distributions: as additional shape combinations are introduced, the model must balance greater intra-class variability with the need for discriminative features, a trade-off that becomes harder to resolve in these richer, noisier datasets. Considering these results, we adopt $N=3$ as default SF-variants and report the full sweeps ($N=1-5$) in the supplementary.

**Detailed Quantitative Comparison.** Table 1 benchmarks SCENEFORGE on the considered zero-shot 3D–text classification suites. For each backbone we report its *best* composition size ($N = 3$) together with its single-shape baseline ($N = 1$), and list recent and sota models for reference.

*Effect of* SCENEFORGE. Across all three backbones SCENEFORGE delivers consistent top-1 gains. With the "no-LVIS" training split the average improvement is $+1.50$ pp for OpenShape, $+0.78$ pp

| Method | 1-shot | | 2-shot | |
|---|---|---|---|---|
| | mIoU | Δ | mIoU | Δ |
| OmniBind-L | 77.2 | | 79.9 | |
| OmniBind-F | 77.8 | – | 80.3 | – |
| OpenShape | 74.0 | | 76.5 | |
| SF-OpenShape | 76.2 | +2.2 | 79.1 | +2.6 |
| ViT-Lens | 75.5 | | 77.9 | |
| SF-ViT-Lens | 77.0 | +1.5 | 80.1 | +2.2 |
| Uni3D | 75.9 | | 78.2 | |
| SF-Uni3D | 78.5 | +2.6 | 81.2 | +3.0 |

Table 2: One-shot and two-shot part segmentation on ShapeNetPart.

| Model | B-4 | ΔB-4 | CIDEr | ΔCIDEr | EM | ΔEM |
|---|---|---|---|---|---|---|
| OmniBind-L + BLIP2-FlanT5 | 8.5 | | 62.9 | | 17.1 | |
| OmniBind-F + BLIP2-FlanT5 | 8.3 | – | 62.1 | – | 17.6 | – |
| OpenShape + BLIP2-FlanT5 | 6.3 | | 54.8 | | 14.1 | |
| SF-OpenShape + BLIP2-FlanT5 | 8.1 | +1.8 | 61.5 | +6.7 | 16.9 | +2.8 |
| ViT-Lens + BLIP2-FlanT5 | 7.2 | | 57.5 | | 15.7 | |
| SF-ViT-Lens + BLIP2-FlanT5 | 8.5 | +1.3 | 63.4 | +5.9 | 17.8 | +2.1 |
| Uni3D + BLIP2-FlanT5 | 7.5 | | 58.3 | | 16.4 | |
| SF-Uni3D + BLIP2-FlanT5 | 10.4 | +2.9 | 66.7 | +8.4 | 20.5 | +4.1 |

Table 3: Performance on the ScanQA dataset using BLEU-4, CIDEr, and Exact Match.

for ViT-Lens and $+1.73$ pp for Uni3D; with the "+LVIS" split the corresponding gains are $+1.43$, $+0.85$ and $+1.75$ pp. Uni3D benefits most, likely a consequence of its larger capacity, which can better exploit the richer intra-sample diversity injected by multi-shape compositions, yet even the smaller OpenShape and the adapter-based ViT-Lens improve.

*Comparison with prior works.* All three SCENEFORGE variants surpass previous non-ensemble methods (ULIP-2 [23], TAMM [30], MixCon3D [8]) on every dataset. Moreover, SF-*Uni3D* outperforms OmniBind [21], the strongest published ensemble, despite using a *single* model: on the LVIS–ModelNet–ScanObjNN–ScanNet quartet it achieves absolute top-1 margins of $+0.7$, $+1.6$, $+0.5$ and $+3.1$ pp, respectively. These results underscore that structured multi-object augmentation offers a more inference-efficient strategy for enhancing representations than costly ensemble methods.

*Why do multi-object compositions enhance single-object classification?* While it might initially appear counterintuitive, training with multi-object compositions fosters improved representations that benefit single-object recognition tasks. This phenomenon aligns with established findings in representation learning literature, particularly in image classification, where multi-sample augmentations (e.g., CutMix, MixUp) are known to induce smoother decision boundaries and promote robust generalization by exposing models to more diverse feature combinations. Analogously, in our structured 3D scene compositions, the increased complexity and relational context implicitly regularize the learned representations, facilitating the emergence of discriminative features resilient to variations in single-object scenarios encountered at inference. By analyzing the positive–negative similarity margins in zero-shot classification, we find that SF-variants yield larger margins over baselines, indicating stronger inter-class separation and more robust decision boundaries. We report the quantitative analysis in the supplementary material.

## 4.2 Few-Shot Part Segmentation on PartNet

We follow the protocol of Uni3D [31] on the ShapeNetPart dataset [25], evaluating each backbone in one-shot and two-shot regimes. As in PointNet++ [18], we freeze the pretrained transformer encoder and attach lightweight feature-propagation heads that upsample intermediate representations to dense part predictions. Only these heads are fine-tuned on the few labeled part annotations.

Table 2 reports mean IoU (mIoU) and the improvement $\Delta$ over the corresponding single-shape baseline for all three backbones (*OpenShape*, *ViT-Lens*, *Uni3D*) with and without SCENEFORGE. SCENEFORGE yields consistent gains: SF-OpenShape improves by +2.2/ +2.6 pp in one-/two-shot, SF-ViT-Lens by +1.5/ +2.2 pp, and SF-Uni3D by +2.6/ +3.0 pp, with SF-Uni3D achieving the highest absolute mIoU (78.5 / 81.2). These improvements suggest that multi-shape pretraining encourages more fine-grained, group-based features that transfer effectively to part segmentation, structuring the feature space to better capture part-level relationships even under extreme label scarcity.

## 4.3 3D Question Answering on ScanQA

We further assess 3D–text alignment on the ScanQA benchmark [2, 9], which requires answering natural-language questions about ScanNet scenes. Following prior works, we freeze each 3D encoder and attach it to BLIP2–FlanT5 [4, 15], then fine-tune on ScanQA's question–answer pairs.

In addition to the sota contrastive baselines (OmniBind-Large, OmniBind-Full), we include SF-OpenShape, SF-ViT-Lens and SF-Uni3D as our multi-shape variants, alongside their single-shape counterparts. We report here BLEU-4 (B-4), CIDEr and Exact Match (EM); full metrics appear in the supplementary. Table 3 gives the results. All three backbones see substantial gains with SCENEFORGE: SF-OpenShape improves B-4 by +1.8 pp, CIDEr by +6.7 pp and EM by +2.8 pp; SF-ViT-Lens adds +1.3 pp, +5.9 pp and +2.1 pp, respectively; and SF-Uni3D leads with +2.9 pp, +8.4 pp and +4.1 pp. While all backbones gain, Uni3D yields the largest relative and absolute boosts, consistent with its greater capacity, whereas ViT-Lens, despite smaller gains, still surpasses OmniBind. A qualitative review shows that, while baseline encoders match our variants on attribute- or color-based questions, SCENEFORGE variants significantly outperform them on spatial reasoning queries, e.g. "What is over the brown chair?", where modeling inter-object relationships is essential. This suggests that our structured multi-shape augmentation not only sharpens local feature representations but also boosts the encoder's ability to infer complex spatial configurations, a critical capability for 3D scene understanding.

## 4.4 Supervised Fine-Tuning

To test if the benefits of SCENEFORGE pre-training extend to supervised settings, we evaluate full fine-tuning and Parameter-Efficient Fine-Tuning (PEFT) methods [10, 16, 32]. We perform experiments on three of the considered classification benchmarks: the synthetic ModelNet40, the challenging real-world ScanObjectNN, and ScanNet Instances extracted from complex indoor scenes.

Table 4: Supervised fine-tuning accuracy (%).

| Model | Method | Trainable Params | ModelNet40 | ScanObjectNN | ScanNet Inst. |
|---|---|---|---|---|---|
| Uni3D | Full Fine-Tuning | 1016.5M (100%) | 94.28 | 97.12 | 82.72 |
| | Adapter | 7.6M (0.74%) | 94.35 | 96.80 | 81.42 |
| | DAPT | 7.3M (0.72%) | 94.33 | 96.78 | 82.65 |
| | PointGST | 4.1M (0.40%) | 94.83 | 97.68 | 83.04 |
| SF-Uni3D | Full Fine-Tuning | 1016.5M (100%) | 94.42 | 97.58 | 83.58 |
| | Adapter | 7.6M (0.74%) | 94.46 | 97.09 | 82.56 |
| | DAPT | 7.3M (0.72%) | 94.49 | 97.15 | 83.46 |
| | PointGST | 4.1M (0.40%) | **94.95** | **98.09** | **84.29** |

As shown in Table 4, our SF-Uni3D backbone provides a superior initialization, consistently outperforming the baseline, especially on the complex ScanNet Instances. A strong synergy with PEFT is also evident: the PointGST method surpasses even full fine-tuning while using just 0.4% of the trainable parameters.

## 4.5 N-Objects cross-modal Retrieval

Following prior work [23, 31], we evaluate cross-modal retrieval on the unseen Objaverse-LVIS dataset, measuring top-k accuracy for both 3D-to-text and text-to-3D retrieval. Models are trained on the full ensemble excluding this set, and performance is assessed via cosine similarity retrieval across embedded samples. Beyond standard single-object retrieval, we aim to analyze how well our models understand increasingly multi-object scenes. To this end, we introduce the N-LVIS benchmark, where *each* shape is composed with $N-1$ additional objects. For clarity, we report results only for the best SF backbone (Uni3D with $N = 1..5$) and compare against OmniBind-F [21]; analogous results for other backbones are provided in the supplementary.

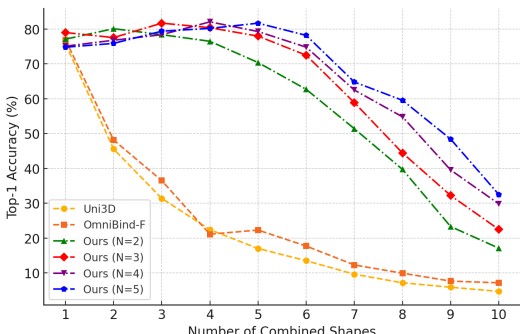

Figure 4: Top-1 averaged retrieval accuracy on the N-LVIS datasets as $N$ increases.

We evaluate retrieval on N-LVIS from $N{=}1$ (standard retrieval) to $N{=}10$, reporting averaged top-1 accuracy (text to 3D and viceversa) in Figure 4. Prior models degrade sharply when faced with multi-object compositions, dropping below 50% at $N{=}2$. In contrast, our models, trained with

varying numbers of composed objects, exhibit strong generalization, each peaking near its training composition size. Notably, the $N=3$ model sustains over 70% accuracy at $N=6$ and around 60% at $N=7$, highlighting its robustness in complex scenes. This comes at a slight cost: models trained with higher $N$ perform marginally worse at lower $N$, reinforcing our findings in Section 4.1 that simpler compositions better preserve single-object understanding. Full top-1 and top-5 results are provided in the supplementary.

## 4.6 Object Repositioning

To qualitatively assess our approach ability to improve reasoning about spatial relationships, we introduce a simple object repositioning task. Given two objects, we combine them in an initial configuration using our MSF. The combined caption is then modified to describe a new spatial relationship, and we optimize a three-parameter offset vector to reposition the second object, maximizing alignment with the updated description while keeping the 3D-text encoder frozen. Full optimisation details appear in the supplementary. As illustrated in Figure 5 encoders

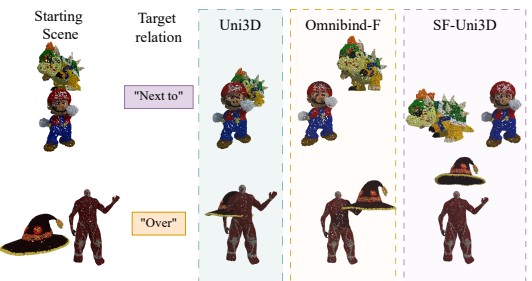

Figure 5: Object repositioning example.

trained with SCENEFORGE reliably relocate the object to the target position, while baseline encoders stall in semantically plausible yet misaligned configurations. We observe the same qualitative behaviour with all other variants, confirming that SCENEFORGE strengthens spatial and compositional reasoning.

# 5 Ablation Studies

## 5.1 Proportion of Multi-Object Samples

The hyperparameter $\alpha$ determines the fraction of multi-object samples in each training batch. We analyze its impact by varying $\alpha$ from 0 to 1 and evaluating the average zero-shot top-1 accuracy on the zero-shot datasets (Figure 6). Increasing $\alpha$ initially improves performance with respect to the baselines, with accuracy peaking around $\alpha = 0.5$. However, for larger $\alpha$, performance declines, likely due to an excessive focus on compositional relationships at the expense of single-object understanding.

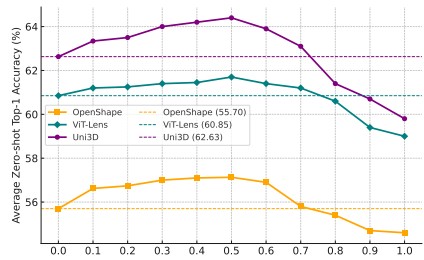

Figure 6: Effect of varying $\alpha$ on average zero-shot top-1 accuracy.

## 5.2 Do other composition functions work?

To assess the impact of our 3D scene composition method, we conduct experiments by fixing the training method to ours, the backbone to the strongest we found (Uni3D) and just ablate the composition method against object composition approaches: Point-CutMix [29] and PointMixUp [3]. PointCutMix replaces regions of one

| Composition Method | Lvis Top-1 | ModelNet Top-1 | ScanObjNN Top-1 | Scannet Top-1 |
|---|---|---|---|---|
| None (Uni3D) | *53.5* | *87.3* | 63.9 | 45.8 |
| PointCutMix-R | *53.5* | 87.1 | *64.1* | *47.5* |
| PointCutMix-K | 44.7 | 83.0 | 45.1 | 34.8 |
| PointMixup | 39.2 | 78.7 | 41.4 | 30.2 |
| SF-Uni3D (N=2) | **53.9** | **87.6** | **64.5** | **48.2** |

Table 5: Different 3D composition methods on zero-shot cls.

point cloud with those from another and has two variants: PointCutMix-R, which randomly replaces points, and PointCutMix-K, which preserves key structures. PointMixUp interpolates point coordinates and features between objects. For a fair comparison, since these methods combine two objects, we compare their results specifically to our $N = 2$ configuration. Textual descriptions for PointCutMix and PointMixUp compositions are generated by concatenating object captions with "and" and subsequently refined using Qwen [24], following our pipeline.

Results in Table 5 show our method consistently outperforming all baselines. Although PointCut-Mix-R exceeds Uni3D on a most benchmarks, it still lags behind our method, even more considering the $N = 3$ variant; PointCutMix-K and PointMixUp perform worse than Uni3D. This gap stems from how each approach handles object semantics and spatial coherence: our method builds structured scenes that align naturally with captions, whereas PointCutMix-R randomly mixes whole objects, creating overlaps, and PointCutMix-K and PointMixUp fragment or interpolate shapes, producing unrealistic, poorly described scenes. Overall, maintaining clear object semantics and spatial relationships through structured composition yields superior generalization.

## 5.3 Are simple relations enough?

To test if the learned spatial understanding generalizes beyond the simple pre-training relations, we isolate performance on ScanQA questions involving more complex, unseen spatial queries. Table 6 shows that our compositionally-trained models consistently outperform their baselines across all backbones and a variety of complex relations.

Table 6: Generalization to unseen spatial relations on ScanQA for all backbones.

| Relation Type | Metric | OpenShape | | | ViT-Lens | | | Uni3D | | |
|---|---|---|---|---|---|---|---|---|---|---|
| | | Baseline | SF | Δ | Baseline | SF | Δ | Baseline | SF | Δ |
| Attached To (21) | CIDEr | 54.5 | 61.5 | +7.0 | 57.1 | 63.3 | +6.2 | 57.9 | **66.6** | +8.7 |
| | EM | 14.1 | 17.1 | +3.0 | 15.6 | 17.9 | +2.3 | 16.5 | **20.8** | +4.3 |
| Sitting On (59) | CIDEr | 56.8 | 63.4 | +6.6 | 59.0 | 65.1 | +6.1 | 61.0 | **70.1** | +9.1 |
| | EM | 15.2 | 17.7 | +2.5 | 16.6 | 18.4 | +1.8 | 17.5 | **22.6** | +5.1 |
| Between (112) | CIDEr | 54.1 | 61.2 | +7.1 | 56.8 | 62.9 | +6.1 | 57.2 | **66.5** | +9.3 |
| | EM | 14.0 | 17.0 | +3.0 | 15.5 | 17.9 | +2.4 | 15.8 | **20.5** | +4.7 |
| Closest To (112) | CIDEr | 55.0 | 61.8 | +6.8 | 57.5 | 63.5 | +6.0 | 58.5 | **67.0** | +8.5 |
| | EM | 14.3 | 17.2 | +2.9 | 15.8 | 18.0 | +2.2 | 16.2 | **20.4** | +4.2 |
| In Front Of (246) | CIDEr | 56.1 | 62.5 | +6.4 | 58.2 | 64.0 | +5.8 | 60.3 | **68.3** | +8.0 |
| | EM | 14.9 | 17.6 | +2.7 | 16.3 | 18.2 | +1.9 | 17.1 | **21.8** | +4.7 |

The comprehensive results in Table 6 show that SCENEFORGE provides consistent benefits across all three backbones. Each SF-variant significantly outperforms its respective baseline on all complex relation types. This indicates that our pre-training builds a robust spatial foundation that generalizes effectively to more nuanced relational queries, regardless of the underlying encoder architecture.

## 6 Limitations and Future Directions

While SCENEFORGE consistently enhances 3D–text alignment across multiple backbones and tasks, we are aware of its limitations. First, our synthetic scene generation employs only three basic spatial relations, which, although diversified through LLM-based refinement, do not fully capture the complexity of natural environments. Future research could focus on more realistic and varied364compositions guided by learned object co-occurrence patterns and spatial priors. Second, due to computational constraints, we maintain a fixed 10 k-point budget for multi-object compositions, resulting in accuracy degradation for densely populated scenes (see Figure 3). Addressing this will require exploring larger point budgets or employing more sophisticated sampling techniques to preserve salient geometric features in complex scenarios. Third, although we leverage the lightweight Qwen2.5 model for refinement, the overhead is not always negligible and reducing it introduces a memory-time tradeoff. Finally, due to rendering costs, we could not pair each composition with a synthetic image, though extending the pipeline to incorporate rendered views for joint 2D–3D learning or studying alternative approaches, such as aligning 3D compositions to aggregations of the single image embeddings, remains a promising direction. Overall, addressing these limitations will significantly broaden the practical impact and robustness of compositional 3D–text learning methods.

**Acknowledgments.** This paper is supported by the PNRR-PE-AI FAIR project funded by the NextGeneration EU program. We acknowledge ISCRA for awarding this project access to the LEONARDO supercomputer, owned by the EuroHPC Joint Undertaking, hosted by CINECA (Italy) .

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
