# OpenReview forum: "SceneForge: Enhancing 3D-text alignment with Structured Scene Compositions"
_NeurIPS.cc/2025/Conference — NeurIPS 2025 poster_

### Official Review · Reviewer_ikuk · 2025-06-20

**Clarity:** 3
**Significance:** 3
**Originality:** 3
**Rating:** 3
**Confidence:** 4

**Summary:**

This paper introduces SCENEFORGE, a novel data augmentation framework designed to enhance contrastive learning alignment between 3D point clouds and text by constructing structured, multi-object 3D scenes. The core idea is to utilize individual 3D object point clouds and combine them into complex scenes based on predefined, explicit spatial relations (e.g., "over," "under," "next to"). Simultaneously, the method leverages a Large Language Model (LLM), specifically Qwen2.5, to "forge" the individual object text descriptions into a single, coherent, and natural scene description that accurately reflects the spatial layout. By mixing these synthetic "compositional" samples with original single-object samples in training batches, SCENEFORGE effectively increases the diversity and complexity of the training data, thereby alleviating the scarcity of large-scale 3D-text datasets.

**Questions:**

1. The success of SCENEFORGE appears to stem from a combination of factors: stronger spatial relationship modeling, more challenging contrastive samples, and richer linguistic descriptions from the LLM. Could you design an experiment to try and disentangle the contributions of these factors? For example, what would the performance be if you used a fixed template (e.g., "A model of [object1] is [relation] a model of [object2]") to generate captions, without refinement from the LLM? This would help isolate the precise contribution of the LLM "forging" process.

2. To better understand the method's boundaries, could you provide an analysis of some failure cases? For instance, are there certain object combinations or spatial relationships that SCENEFORGE struggles to generate or align correctly? Or, in what situations does the LLM produce erroneous or ambiguous scene descriptions?

3. In the methods section, you mention that the "next to" placement is achieved by sampling a horizontal unit vector d. How is this vector d sampled? Is it re-sampled randomly for each placement, or is it consistent for all "next to" relationships within a single composed scene? This detail could affect the naturalness of the scene layouts.

4. As noted in the paper, the fixed point budget and simple spatial relations are current limitations. What do you believe is the best path forward to address this? Is it simply a matter of increasing the point budget (e.g., having it scale dynamically with the number of objects N), or are more intelligent point cloud sampling strategies needed to preserve salient geometric features? On the spatial relations front, how could a richer set of relations be introduced in a cost-effective manner?

5. The experiments show that α=0.5 is an optimal choice. This value seems intuitive (half single samples, half composed samples). Do you believe this optimal value is universal? Might it change depending on the capacity of the backbone model, the characteristics of the dataset, or the number of combined objects N?

**Ethical Concerns:**

["NO or VERY MINOR ethics concerns only"]

**Final Justification:**

Thank authors for addressing my previous concerns. But I think the technical contribution of this work is still rather limited. I maintain my original score.

**Limitations:**

yes

**Quality:**

2

**Strengths And Weaknesses:**

Strengths:
1. While most contemporary work focuses on improving model architectures, SCENEFORGE takes a unique, data-centric approach, proposing a model-agnostic, plug-and-play framework. Through the "scene forging" process, it not only virtually expands the dataset but, more importantly, introduces "harder samples" with structured and relational information, which is crucial for enhancing a model's generalization and spatial reasoning capabilities.
2. The authors have conducted exceptionally thorough and rigorous experiments to validate their method's efficacy.
3. Not only does SCENEFORGE achieve consistent gains over all baselines, but its SF-Uni3D variant also outperforms the powerful ensemble model OmniBind on several zero-shot classification benchmarks.

Weaknesses:
While the results presented in this paper are strong, I have significant reservations about the novelty of the core methodology. At its heart, the proposed method, SCENEFORGE, is a direct data augmentation technique. It does not introduce a new model architecture or a fundamental change to the contrastive learning framework itself.
The contribution is primarily at the data level—synthesizing more complex and diverse training samples by composing existing ones. Although this is executed cleverly with the use of an LLM for caption refinement, the overall approach can be framed as an advanced form of data engineering rather than a novel method in representation learning. The work, despite its effectiveness, may be perceived as incremental from a methodological standpoint.

---

> ### Author Rebuttal · Authors · 2025-07-28
>
> We thank the reviewer for their detailed analysis and for recognizing our "exceptionally thorough and rigorous experiments." We appreciate the opportunity to clarify why SCENEFORGE is a methodological contribution, not an incremental augmentation.
>
> We agree that our work is data-centric, and we argue this is precisely where its methodological novelty lies. In a field where architectural improvements can yield diminishing returns, we demonstrate that innovating on the learning signal itself is a powerful and novel frontier. Here's why:
>
> - Data-Centric Novelty: The novelty of SCENEFORGE lies in its structured manipulation of the input-label space, a powerful way to regularize the model and improve its representations. This approach of innovating on the learning signal itself, rather than the architecture, is precisely why foundational 2D methods like MixUp and CutMix were considered novel. SCENEFORGE introduces a similar principle to the 3D-text domain, but goes further by composing scenes that preserve object semantics while injecting new, structured relational information.
>
> - Fundamentally Alters the Learning Task: We shift the objective from only object recognition to a more complex task including contextual scene understanding. By introducing in the batch structured 3D scenes with LLM-refined scene captions, we force the model to learn spatial relationships and part-whole reasoning. Our experiments prove this is crucial for enhancing the model's spatial reasoning and generalization.
>
> - Structured Composition, Not Direct Mixing: Unlike direct augmentations like PointMixup, our structured forging process ensures geometric and semantic coherence. Our ablation study (Table 4) proves this structured approach is the key to our performance.
>
> - Novelty Validated by Impact: Our framework's impact is the strongest evidence of its novelty. The fact that our single model outperforms a powerful SOTA ensemble (OmniBind) shows that a better learning signal can be more impactful than adding architectural complexity.
>
> We hope this clarifies our perspective. We believe this reframes SCENEFORGE as a novel methodological contribution, and we will gladly revise the manuscript to emphasize these points more clearly.
>
> ## Questions
> Q1: LLM ablation
>
> We agree that ablating the LLM component in the pipeline is crucial. We ran this exact experiment (Supp. Table 2): we compare our full method (using Qwen 2.5) against a variant that uses raw, template-based captions without any LLM refinement (SF-Uni3D/NO LLM). These raw captions are constructed by simply concatenating object descriptions with spatial relations (e.g., "[object1] next to [object2]"), which is analogous to the reviewer's proposed template-based approach. The "NO LLM" variant, which uses only structured 3D scenes and template-based text, already shows clear improvements over the Uni3D baseline across all benchmarks. However, refining these raw captions with an LLM provides a second, significant boost in performance. We also report and observe similar boost also with LLama 3.1 instead of Qwen 2.5.
>
> Q2: Failure cases.
>
> On the point-cloud composition side, our choices of positional relationship and the method we propose to apply them ensures us that no failure cases can occurr. For the LLM side, there are possible cases of hallucination and information alteration. To analyze LLM failure cases, following the reviewer insightful question, we built an automated verification pipeline using dependency parsing with spaCy (classic NLP). This tool programmatically extracts subjects (nouns) and their descriptive attributes (adjectives), and we do that for both the raw concatenated captions and the final refined captions. We ran this check on 1M random combined scene captions with SCENEFORGE, and we found that in over 98% of cases, original subjects and their attributes were perfectly preserved. In the ~2% failure cases, 79% of them are for wrong attributes (attribute binding is indeed a very well known problem of LLMs), 20% are missing attributes and just 1% is about wrong subjects. Even with this unfrequent possible failure cases, the use of the LLM for rewriting brings crucial benefits to the pipeline, as discussed in the previous point.
>
> Q3: d vector
>
> d in the xy-plane is re-sampled randomly for each "next to" placement, and it is identified by uniform sampling its angle wrt x axis. Inside single scenes, we only check for intersections with previous placed objects in case of #objects > 2 to avoid unrealistic scenes, and in that case we resample it. This allows us to produce all valid and variegate positionings in the considered potential scene.
>
> Q4: Fixed point budget & spatial relations
> - On Point Budget and Sampling Strategies:
> While re-running our full set of experiments with larger point budgets was (and is) unfortunately outside our computational budget, we are happy to share results from some preliminary experiments conducted for this paper that explore this exact question up to 20k points per shape. It's important to note that these tests were run on a 20x smaller training dataset, a 50k random subset of Objaverse (no LVIS labels) on the Uni3D variant.
>
>     ##### LVIS
>     |Pts.|N=1|N=2|N=3|N=4|N=5|
>     |:---|:----|:----|:----|:----|:----|
>     |**5k**|27.24|**27.59**|27.26|27.02|26.58|
>     |**10k**|27.41|28.06|**28.15**|27.64|27.33|
>     |**15k**|27.38|28.03|**28.18**|28.09|27.79|
>     |**20k**|27.47|28.10|**28.13**|**28.13**|28.10|
>
>     ##### ModelNet
>     |Pts.|N=1|N=2|N=3|N=4|N=5|
>     |:---|:----|:----|:----|:----|:----|
>     |**5k**|65.04|**65.51**|65.11|64.27|63.89|
>     |**10k**|65.12|65.83|**66.28**|65.51|65.04|
>     |**15k**|65.17|65.85|**66.26**|66.21|65.58|
>     |**20k**|65.15|66.04|66.25|66.25|**66.28**|
>
>     ##### ScanObjNN
>     |Pts.|N=1|N=2|N=3|N=4|N=5|
>     |:---|:----|:----|:----|:----|:----|
>     |**5k**|48.98|**49.53**|48.80|48.52|47.96|
>     |**10k**|49.03|49.71|**50.01**|49.45|49.08|
>     |**15k**|49.10|49.80|50.04|**50.11**|49.68|
>     |**20k**|49.08|49.92|**50.09**|50.05|**50.09**|
>
>     ##### ScanNet
>     |Pts.|N=1|N=2|N=3|N=4|N=5|
>     |:---|:----|:----|:----|:----|:----|
>     |**5k**|25.98|**26.47**|26.25|25.04|24.79|
>     |**10k**|26.17|27.23|**27.45**|26.81|26.10|
>     |**15k**|26.15|27.22|**27.48**|27.44|27.09|
>     |**20k**|26.21|27.28|27.44|**27.47**|27.42|
>
>     These results confirm our hypothesis from the paper. The benefit of composition peaks around N=3 objects, aligning with our main findings. More importantly, increasing the point budget primarily makes our model more resilient to complexity, significantly alleviating the performance degradation at N=4 and N=5. This demonstrates that the drop is a direct result of feature fragmentation under a constrained budget, not a fundamental flaw in our compositional method. Therefore, while scaling the point budget is one solution, we agree with the reviewer that testing with more intelligent sampling strategies to preserve salient features is a promising future direction to balance the point budget and accuracy.
>
>
> - Spatial Relations:
> We chose simple spatial relations to ensure our method remains scalable and applicable to the large, uncurated datasets crucial for contrastive learning. Implementing more complex relations (e.g., "inside," "leaning on") would require detailed, object-specific semantic and geometric knowledge (like identifying container properties or support surfaces), not available in large datasets.
>
>     The ideal path forward should seek to introduce these harder, semantic-aware relations without sacrificing scalability. This could involve future work on methods to automatically annotate large datasets with the necessary geometric and semantic priors to enable more complex compositions.
>
>     While this is a promising future direction, our current simple relations already provide a powerful learning signal that generalizes well. A new analysis on the ScanQA dataset confirms this: our method significantly improves performance on unseen, complex relations, proving our pre-training builds a strong, generalizable foundation:
>
>     |Relation Type (num questions)|Metric|OpenShape|SF-OpenShape|Δ|ViT-Lens|SF-ViT-Lens|Δ|Uni3D|SF-Uni3D|Δ|
>     |:----------------------------|:----:|:-------:|:----------:|:-:|:------:|:---------:|:-:|:---:|:---:|:---:|
>     |**AttachedTo** (21)|CIDEr|54.5|61.5|+7.0|57.1|63.3|+6.2|57.9|66.6|+8.7|
>     ||EM|14.1|17.1|+3.0|15.6|17.9|+2.3|16.5|20.8|+4.3|
>     |**SittingOn** (59)|CIDEr|56.8|63.4|+6.6|59.0|65.1|+6.1|61.0|70.1|+9.1|
>     ||EM|15.2|17.7|+2.5|16.6|18.4|+1.8|17.5|22.6|+5.1|
>     |**ClosestTo** (112)|CIDEr|55.0|61.8|+6.8|57.5|63.5|+6.0|58.5|67.0|+8.5|
>     ||EM|14.3|17.2|+2.9|15.8|18.0|+2.2|16.2|20.4|+4.2|
>
> Q5: Value of alpha
>
> Our choice of α=0.5 was based on an empirical sweep with a 0.1 step for each of our three backbones, a granularity dictated by the high computational cost of these experiments.
>
> As shown in Figure 6 of the main paper, we observed that performance increases up to 0.5, but however doesn't hinge on a single sharp peak, but rather forms a plateau of high accuracy between roughly α=0.3 and α=0.6. Interestingly, this trend and the peak performance at α=0.5 were remarkably consistent across all three models, from the adapter-based ViT-Lens to the large-capacity Uni3D. This suggests that the optimal mixture ratio may be largely independent of the model architecture and capacity itself. While the exact global optimum is likely around 0.5, we agree it could still depend on other factors, such as the dataset's intrinsic complexity and, as the reviewer suggests, the maximum number of combined objects (N), a valuable direction to analyze with sufficient compute.
>
>
> We are grateful to the reviewer for their thorough analysis and for pushing us to better articulate the novelty of our data-centric approach. We trust that our detailed response and the new supporting evidence provide a clearer and more compelling case for our contributions and address the initial concerns.

---

> > ### Comment · Reviewer_ikuk · 2025-08-05
> >
> > Thank you for addressing my previous concerns.
> >  But I think the technical contribution of this work is still rather limited.
> > I maintain my original score.

---

> ### Author Response · Authors · 2025-08-05
>
> Thank you for your comment and for your engagement in the discussion process.
>
> We appreciate your acknowledgment that our rebuttal addressed all your previous concerns. We also want to reiterate our gratitude for your honest recognition of the strengths of our work in your review.
>
> We understand that the only remaining point of disagreement is a high-level one regarding the nature of a contribution. As we elaborated in our rebuttal, our work is based on the principle that innovating on the learning signal itself, by means of a structured framework that introduces harder, relationally aware samples, constitutes a significant and novel methodological advancement. We believe that showing that such a data-centric approach can be more impactful than purely architectural or technical changes is a relevant and timely message for the research community. Although we have different views on this core principle, we respect your perspective and final assessment.
>
> Thank you again for your time and thoughtful feedback.

---

### Official Review · Reviewer_Kxh9 · 2025-06-28

**Clarity:** 4
**Significance:** 3
**Originality:** 4
**Rating:** 4
**Confidence:** 3

**Summary:**

This paper proposed SceneForge, which leverages individual 3D shapes to construct multiobject scenes with explicit spatial relations, and train a 3D-text contrastive learning model to pairing them with coherent multi-object descriptions refined by a large language model. This approach boosts performance across several benchmarks.

**Questions:**

1. Is it possible to include more spatial relationships and construct a more complex scene for training?
2. Refer to point 2 of Weaknesses.

**Ethical Concerns:**

["NO or VERY MINOR ethics concerns only"]

**Final Justification:**

The authors' responses addressed my concerns well. Based on the general novelty and contributions, I keep the original rating of borderline accept.

**Limitations:**

yes

**Quality:**

3

**Strengths And Weaknesses:**

Strengths:
1. The ideas are simple but effective. This method delivers consistent gains across multiple 3D-text alignment benchmarks.

Weaknesses:
1. The set of predefined spatial relations is too limited, so it fails to capture more complex or subtle spatial arrangements.
2. Using a fixed 10k point sampling budget causes performance to drop sharply when four or more objects appear. It would help to see results with more points such as 10k, 20k, and 40k points to understand how sampling affects accuracy.

---

> ### Author Rebuttal · Authors · 2025-07-26
>
> We sincerely thank the reviewer for their valuable feedback and positive assessment, especially for recognizing that our ideas are simple but effective and deliver consistent gains. We appreciate the opportunity to clarify the points raised regarding the spatial relations and point budget.
>
> We will now address reviewer highlighted weaknesses and questions, grouped by topic.
>
> ### W1 and Q1:
>
> The reviewer rightly points out that our set of spatial relations is simple and doesn't capture the full complexity of real-world arrangements. This was a deliberate and strategic design choice, and we'd like to elaborate on the reasoning.
>
> Our primary goal was to create a robust, scalable, and model-agnostic augmentation framework. Implementing more complex relations (e.g., "inside", "leaning on") would require detailed, object-specific semantic and geometric knowledge (e.g., identifying container properties or specific support surfaces). Enforcing such constraints would limit our method to smaller, heavily annotated datasets and prevent us from leveraging the scale and diversity of massive, less-curated collections like Objaverse, which is a cornerstone of our approach's success. Our simple relations are universal and object-agnostic, ensuring maximum scalability.
>
> Despite their simplicity, our experiments show that these foundational relations provide a powerful learning signal. The significant improvements in downstream tasks that require nuanced spatial understanding, such as 3D Visual Question Answering, confirm that the model effectively learns useful bases for spatial reasoning even from these basic cues, which form a stronger starting point for finetuning on downstream tasks/using the model as a frozen feature extractor.
> To empirically analyze this last observation, we grouped performance on ScanQA questions involving more complex spatial relations not used during the pretraining of the contrastive model. The results below demonstrate a consistent and significant performance uplift with respect to the non-SCENEFORGE models:
>
> | Relation Type (num questions) | Metric | OpenShape | SF-OpenShape | Δ | ViT-Lens | SF-ViT-Lens | Δ | Uni3D | SF-Uni3D | Δ |
> | :---------------------------- | :----: | :-------: | :----------: | :-: | :------: | :---------: | :-: | :---: | :---: | :---: |
> | **Attached To** (21)          | CIDEr  |   54.5    |     61.5     | +7.0|   57.1   |    63.3     | +6.2| 57.9  | 66.6  | +8.7  |
> |                               | EM     |   14.1    |     17.1     | +3.0|   15.6   |    17.9     | +2.3| 16.5  | 20.8  | +4.3  |
> | **Sitting On** (59)           | CIDEr  |   56.8    |     63.4     | +6.6|   59.0   |    65.1     | +6.1| 61.0  | 70.1  | +9.1  |
> |                               | EM     |   15.2    |     17.7     | +2.5|   16.6   |    18.4     | +1.8| 17.5  | 22.6  | +5.1  |
> | **Between** (112)             | CIDEr  |   54.1    |     61.2     | +7.1|   56.8   |    62.9     | +6.1| 57.2  | 66.5  | +9.3  |
> |                               | EM     |   14.0    |     17.0     | +3.0|   15.5   |    17.9     | +2.4| 15.8  | 20.5  | +4.7  |
> | **Closest To** (112)          | CIDEr  |   55.0    |     61.8     | +6.8|   57.5   |    63.5     | +6.0| 58.5  | 67.0  | +8.5  |
> |                               | EM     |   14.3    |     17.2     | +2.9|   15.8   |    18.0     | +2.2| 16.2  | 20.4  | +4.2  |
> | **In Front Of** (246)         | CIDEr  |   56.1    |     62.5     | +6.4|   58.2   |    64.0     | +5.8| 60.3  | 68.3  | +8.0  |
> |                               | EM     |   14.9    |     17.6     | +2.7|   16.3   |    18.2     | +1.9| 17.1  | 21.8  | +4.7  |
>
> Upon acceptance, we will add this insightful extended analysis to the supplementary material, thanks again to the reviewer for the question.
>
> Future Work: Even if our approach has good generalization to some more complex unseen relations, extending the set of relations is a crucial and interesting direction for future work and practical deployments. As we discuss in the limitations section of our paper , we believe that developing methods for exploiting both large uncurated data and more complex, semantically guided compositions is a promising next step that can build upon the general-purpose foundation we have established here.
>
> W2 and Q2:
>
> On the Fixed Point Sampling Budget: we agree that the relationship between the point budget and scene complexity is a fascinating and important aspect of our work.
>
> While re-running our full set of experiments with larger point budgets was (and is) unfortunately outside our computational budget, we are happy to share results from some preliminary experiments conducted for this paper that explore this exact question. It's important to note that these tests were run on a 20x smaller training dataset, a 50k random subset of Objaverse (without LVIS labels), but we believe the findings directly address the reviewer's curiosity and provide valuable insight.
>
> The table below shows zero-shot accuracy on the manuscript evaluation datasets, varying both the point resolution (rows) and the maximum number of composed objects N (columns). All other hyperparameters are fixed.
>
>
> ##### LVIS
> | Pts. |N=1 | N=2 | N=3 | N=4 | N=5 |
> | :--- | :---- | :---- | :---- | :---- | :---- |
> | **5k** | 27.24 | **27.59** | 27.26 | 27.02 | 26.58 |
> | **10k**| 27.41 | 28.06 | **28.15** | 27.64 | 27.33 |
> | **15k**| 27.38 | 28.03 | **28.18** | 28.09 | 27.79 |
> | **20k**| 27.47 | 28.10 | **28.13** | **28.13** | 28.10 |
>
> ##### ModelNet
> | Pts. |N=1 | N=2 | N=3 | N=4 | N=5 |
> | :--- | :---- | :---- | :---- | :---- | :---- |
> | **5k** | 65.04 | **65.51**| 65.11 | 64.27 | 63.89 |
> | **10k**| 65.12 | 65.83 | **66.28**| 65.51 | 65.04 |
> | **15k**| 65.17 | 65.85 | **66.26** | 66.21| 65.58 |
> | **20k**| 65.15 | 66.04 | 66.25 | 66.25 | **66.28** |
>
> ##### ScanObjNN
> | Pts. |N=1 | N=2 | N=3 | N=4 | N=5 |
> | :--- | :---- | :---- | :---- | :---- | :---- |
> | **5k** | 48.98 | **49.53**| 48.80 | 48.52 | 47.96 |
> | **10k**| 49.03 | 49.71 | **50.01**| 49.45 | 49.08 |
> | **15k**| 49.10 | 49.80 | 50.04 | **50.11**| 49.68 |
> | **20k**| 49.08 | 49.92 | **50.09** | 50.05 | **50.09** |
>
> ##### ScanNet
> | Pts. |N=1 | N=2 | N=3 | N=4 | N=5 |
> | :--- | :---- | :---- | :---- | :---- | :---- |
> | **5k** | 25.98 | **26.47**| 26.25 | 25.04 | 24.79 |
> | **10k**| 26.17 | 27.23 | **27.45**| 26.81 | 26.10 |
> | **15k**| 26.15 | 27.22 | **27.48** | 27.44| 27.09 |
> | **20k**| 26.21 | 27.28 | 27.44 | **27.47** | 27.42 |
>
> The benefit of compositions is evident already at N=2 and plateaus around N=3. The most significant performance gains are achieved when composing 2 or 3 objects. This aligns perfectly with our paper's finding that N=3 is the optimal configuration for the 10k point budget used in our main experiments. Beyond this point, the returns diminish, suggesting a natural plateau. A higher point budget primarily alleviates degradation. This is also in line with our hypothesis that heavy performance drops at high N are due to point budget limitations. However, as the resolution increases from 10k to 20k, the primary effect is not a dramatic increase in peak accuracy. Instead, the larger budget makes our model resilient to training sample complexity, alleviating the performance degradation for N=4 and N=5. This confirms that the performance drop is not a failing of the compositional method itself, but a direct result of feature fragmentation under a constrained point budget.
>
> We trust that our detailed justification for the chosen spatial relations, combined with the new quantitative results showing generalization to complex, unseen relations and the preliminary study on point budgets, have fully addressed the reviewer concerns and questions. We are confident these additions will significantly strengthen the paper and would be grateful if the reviewer would consider this in their final assessment. We once again thank the reviewer for their valuable and constructive feedback.

---

> > ### Comment · Reviewer_Kxh9 · 2025-08-04
> >
> > Thank you for the authors' responses, they addressed my concerns well. I will include all the rebuttals in the considerations for the final ratings.

---

> ### Author Response · Authors · 2025-08-04
>
> Dear Reviewer,
>
> Thank you for your message and for the positive confirmation. We are very glad to hear that our responses successfully addressed your concerns.
>
> We are grateful that you are taking our rebuttal into consideration. If there is any further information or explicit clarification we can provide that would be helpful for your final assessment, we would be more than happy to do so.
>
> Thank you again for your constructive feedback, we look forward to your final recommendation.

---

> > ### Author Response · Authors · 2025-08-07
> >
> > Dear Reviewer,
> >
> > Thank you again for your positive engagement and for confirming that our rebuttal addressed your concerns.
> >
> > As the discussion period is ending in two days, we just wanted to do a final check to see if you had any additional questions or if there was anything we could provide to help with your final assessment.
> >
> > We also wanted to briefly highlight that we have added an extensive new set of experiments on supervised fine-tuning to the public discussion thread. We believe these results provide significant additional evidence for our method's effectiveness, and we are happy to answer any questions you may have about them as well.
> >
> > Thank you once more for your support, valuable time and consideration.

---

### Official Review · Reviewer_3sZR · 2025-07-02

**Clarity:** 4
**Significance:** 4
**Originality:** 3
**Rating:** 5
**Confidence:** 3

**Summary:**

This work augments 3D datasets used for spatial tasks by composing scenes with 3D assets. They show that doing this augmentation on data helps improve model performance on a range of different tasks.

**Questions:**

How are the relations + objects sampled? Uniformly? Do they have to semantically plausible, or does the plausibility of the final scene not matter? I would imagine for downstream applications that require more semantic-level reasoning, that training on semantically implausible scenes would be a problem. Do you observe that in your experiments?

**Ethical Concerns:**

["NO or VERY MINOR ethics concerns only"]

**Final Justification:**

The issues raised in my review was adequately addressed by the rebuttal, and as such I maintain  the positive rating.

**Limitations:**

Yes

**Quality:**

4

**Strengths And Weaknesses:**

+ Strength

  The method is shown to consistently improve performances including zero-shot classification,  retrieval, segmentation and VQA. Therefore, the demonstrable impact of this work is quite large.

  Additionally, because the method is a way of augmenting 3D data, it is also model agnostic.

  The  analysis of design choices is thorough.

+ Weaknesses

  The scheme described in "3D Scene Forge" ( Section 3.1 ) leads me to think that the kind of final placements that can be synthesized would be quite simplistic, and likely would not take into account the fine-grained geometry of either object (e.g. surfaces on shelves)


  Also, how were the rendering views chosen?

---

> ### Author Rebuttal · Authors · 2025-07-26
>
> We are deeply grateful to the reviewer for their positive and encouraging feedback, their high rating of our work, and their insightful questions. We are glad they found our method's impact to be large, model-agnostic, and our analysis thorough. We will address the points raised below.
>
> We agree with the reviewer that our current placement strategy is a simplified abstraction of real-world interactions and does not explicitly model fine-grained geometry like placing an object on a shelf. This was a deliberate design choice aimed at creating a robust, scalable, and general-purpose augmentation method. Implementing more complex relations like "laying on top of" or "inside" would require detailed, object-specific knowledge. For example, "laying on" would necessitate identifying suitable planar surfaces on the target object, while "inside" would require semantic knowledge about which objects can be containers.
>
> Enforcing such constraints would have severely limited our compositions or training data itself to smaller, highly-curated datasets where this information is available. This would forfeit the primary advantage of leveraging massive, diverse, and less-curated datasets like Objaverse, which is critical for learning generalizable representations.
>
> Therefore, we opted for a small set of universal and object-agnostic spatial relations. These simple relations can be applied to any pair of objects without prior knowledge, ensuring maximum scalability. Despite their simplicity, they are foundational for spatial understanding and, as our results demonstrate, provide a powerful training signal. Our strong performance on the ScanQA VQA benchmark and the object repositioning task  confirm that the model effectively learns meaningful spatial reasoning from these cues. We see this as a foundational step. As mentioned in our Limitations section (Sec. 6), incorporating more sophisticated, semantically-aware priors is a promising future direction that can build upon the strong, general-purpose base established by our current approach.
>
>
> We thank again the reviewer: in the camera ready version, we will make more clear the just explained motivations behind our dseign choices for the considered positional relations.
> ## Questions
> - How are relations + objects sampled? As highlighted above, thanks to the universal applicability of our relations, we can sample both the additional objects and the spatial relations uniformly at random from the training set and our predefined relation set, respectively. In each scene, objects are sampled without replacement to avoid ambiguities, while relations are sampled with replacement to allow a greater variability.
>
>
> - About plausibility: the reviewer rightly notes that this random sampling can lead to semantically implausible scenes. We believe indeed that even implausible scenes can be helpful. By exposing the model to a vast and diverse set of combinations, both plausible and implausible, we push it to learn disentangled and robust features for individual objects. To understand an unlikely scene, the model must first develop a strong, independent understanding of each component. This process is analogous to aspects of human learning. We can easily imagine fantastical or absurd scenes, and this ability to compose known concepts in novel ways strengthens, rather than weakens, our understanding of the concepts themselves. Moreover, filtering unplausible scenes would require additional knowledge or limiting the available data, which falls into the same data quality-scale tradeoff described above.
>
>     Furthermore, it is crucial to distinguish between the types of "implausibility" our method generates versus the ones we discussed above with complex relations:
>
>     - Our method creates scenes that may be semantically implausible but remain geometrically coherent. For example, a "a planet next to a dinosaur" is unlikely but physically possible to arrange. The spatial relationship is simple and universally applicable.
>
>     - We instead avoid creating scenes that are geometrically nonsensical, such as " a planet inside a dinosaur." This would require complex, object-specific modifications (like cutting a hole) that would break the geometric integrity, and possibly semantic properties, of the shapes and create a confusing learning signal to align with the caption.
>
> - Did we notice problems with implausible scenes? Our experiments indicate that our training strategy is beneficial: when the encoder is used as a feature extractor for downstream tasks requiring strong semantic reasoning, such as ScanQA evaluation, our method consistently outperforms the baseline. We did perform a qualitative review of failure cases in the VQA task, but we did not find any significant pattern suggesting that training on implausible scenes led to specific types of errors in real data.
>
> - For Image renderings, we stick with the approach from the baseline model Uni3D for fairness, rendering 10 color images from different views that uniformly cover the whole shape, then we uniformly sample one of 10 each time during training steps.
>
> Thank you once again for your valuable feedback and strong support. We hope these clarifications have fully addressed your questions.

---

> > ### Comment · Reviewer_3sZR · 2025-08-09
> >
> > I'd like to thank the authors for your in depth rebuttal --  as I was already rating positively on this work, I will maintain that rating.

---

### Official Review · Reviewer_1m9F · 2025-07-03

**Clarity:** 3
**Significance:** 3
**Originality:** 3
**Rating:** 5
**Confidence:** 4

**Summary:**

This paper proposes SCENEFORGE, a compositional data augmentation pipeline for 3D vision-language pretraining. The method dynamically synthesizes multi-object 3D scenes and generates corresponding structured text descriptions via large language models, aiming to enhance 3D-text alignment and downstream performance in tasks such as part segmentation and scene classification. Extensive experiments demonstrate consistent improvements across multiple benchmarks and backbones.

**Questions:**

1. For the ScanNet experiments, is instance classification performed on extracted object instances or entire scenes? Please detail the data processing and evaluation protocol.
2. Which variant of ScanObjectNN is used in the experiments (e.g., PB_T50_RS)? Please specify, as this affects comparability with prior work.

**Ethical Concerns:**

["NO or VERY MINOR ethics concerns only"]

**Final Justification:**

I suggest raising the score to 5, pending the authors' inclusion of the discussion in the final version.

**Limitations:**

yes

**Quality:**

3

**Strengths And Weaknesses:**

Strengths：
1. The method is general and can be integrated with existing 3D encoders.
2. Experimental results are comprehensive, covering several datasets and tasks, with clear improvements over baselines.

Weaknesses：
1. The paper lacks quantitative analysis of the latency and resource overhead introduced by online data synthesis, which is important for practical deployment.
2. The mechanism by which compositional scene structure benefits part segmentation and classification tasks is not sufficiently explained.

---

> ### Author Rebuttal · Authors · 2025-07-28
>
> We thank the reviewer for their insightful feedback and constructive suggestions. We are encouraged that the reviewer found our method to be general, our experimental results comprehensive, and the improvements clear. We will address the weaknesses and questions raised point-by-point.
>
> ### W1: Latency and Resource Overhead
> We provide a detailed analysis of latency and resource overhead in the supplementary material (Sec. 5), for which we briefly report the results in the main document (Sec. 3.2). We will reference the supplementary analysis more prominently in the main paper upon acceptance. To summarize the key findings for the reviewer's convenience:
>
> First, our method introduces no practical deployment latency: at inference time, the computational cost is identical to the baseline model, as the model architecture and forward pass are unchanged.
>
> The additional computational overhead from our method is confined entirely to the training stage as a one-time, offline cost. The point cloud combination cost is negligible, as it is composed just by fast operations as uniform sampling of samples and positional relations, and then 3D transformations and array concatenations. Instead, the LLM-based caption refinement does introduce an overhead, which however we show can be effectively eliminated (Supplementary Sec. 5). For instance, by using quantized LLM replicas on a single GPU, the training overhead is removed even for the fastest backbones. Therefore, our method delivers significant performance gains without compromising the efficiency required for practical deployment.
>
> ### W2: Benefits for part segmentation and classification
> We thank the reviewer for highlighting this aspect, as it allows us to further clarify the mechanisms behind the performance improvements. We will address the benefits for part segmentation and single-object classification separately.
>
> - On the Benefit for Part Segmentation.
>
>     Our hypothesis is that compositional training encourages the model to learn more fine-grained, part-aware features, which serve as a superior starting point for the few-shot part segmentation task. To correctly interpret composite scenes which change at every sampling (both in subjects and relation), the model is discouraged from simply learning a holistic representation. It it pushed instead to first learn to distinguish the constituent objects ("hat," "person") before it can reason about their spatial relationship ("over"). This process implicitly forces the 3D encoder to develop more localized and semantically meaningful features that can isolate object parts. Because the downstream part-segmentation task is performed in a few-shot setting, the quality of the pretrained encoder is critical. By pretraining on our compositional scenes, the encoder is already primed to understand part-level relationships, providing a much stronger feature representation for the lightweight segmentation heads to fine-tune from. The consistent gains shown in Table 2 of our main paper support this conclusion.
>
> - On the Benefit for Single-Object Classification.
>
>     Our approach is directly inspired by a well-established principle from representation learning literature: multi-sample augmentations act as powerful regularizers. In 2D vision, techniques like MixUp and CutMix are proven to enhance model robustness and generalization. They do so by creating diverse, challenging training samples that encourage the learning of smoother decision boundaries. Just as MixUp and CutMix pair their augmented images with corresponding soft labels, our method pairs its composite 3D scenes with rich, LLM-refined textual scene descriptions. This descriptive caption effectively functions as a semantic soft label. Instead of merely interpolating class probabilities, it weaves together multiple distinct concepts and their spatial relations into a single, refined target description. This parallel is particularly potent in our zero-shot context, where text prompts are the labels at inference time. By training the encoder to align with these complex "semantic soft labels," we directly encourage a more organized and discriminative feature space. Ultimately, this structured composition of both the 3D input and the textual "label" forces the model to understand each object's features within a complex relational context, yielding a more robust representation that generalizes effectively to the simpler task of single-object classification.
>
>     To provide direct quantitative evidence for this, we did analyze the cosine margin, a standard metric for evaluating the quality of a learned representation space. The margin is defined as the difference between the cosine similarity of a sample to its correct (positive) class and the similarity to its most confusing incorrect (hardest negative) class. A larger margin indicates better class separation and a more robust model.
>
>     The results below show that our SCENEFORGE (SF) method consistently and significantly improves this margin across all three backbones and all four benchmark datasets.
>
>     | Dataset | Model | Baseline Margin | SF Margin | **Δ Margin** |
>     | :--- | :--- | :---: | :---: | :---: |
>     | **ModelNet** | Uni3D | 0.070 | 0.094 | **+0.024** |
>     | | ViT-Lens | 0.060 | 0.075 | **+0.015** |
>     | | OpenShape | 0.045 | 0.065 | **+0.020** |
>     | --- | --- | --- | --- | --- |
>     | **LVIS** | Uni3D | -0.042 | 0.003 | **+0.045** |
>     | | ViT-Lens | -0.045 | -0.015 | **+0.030** |
>     | | OpenShape | -0.070 | -0.030 | **+0.040** |
>     | --- | --- | --- | --- | --- |
>     | **ScanObjNN** | Uni3D | 0.023 | 0.046 | **+0.023** |
>     | | ViT-Lens | 0.018 | 0.030 | **+0.012** |
>     | | OpenShape | 0.010 | 0.028 | **+0.018** |
>     | --- | --- | --- | --- | --- |
>     | **ScanNet** | Uni3D | -0.061 | -0.023 | **+0.038** |
>     | | ViT-Lens | -0.050 | -0.032 | **+0.018** |
>     | | OpenShape | -0.085 | -0.055 | **+0.030** |
>
>     Crucially, on the most challenging datasets (LVIS and ScanNet), where baselines often exhibit a negative margin, meaning they are more confident in an incorrect class than the correct one, our method provides the largest improvements. For example, SF-Uni3D pushes the LVIS margin from negative to positive. This demonstrates that our compositional training directly enhances the model's ability to discriminate between fine-grained or noisy objects, confirming that it yields a more robust and better-organized representation space.
>
>
>     We thank the reviewer for the insightful question: upon acceptance, we will add these results, as it was our intention, to the supplementary material. We will also improve the explaination of the mechanisms behind the benefits in the already dedicated main document section.
>
> ## Questions
> - ScanNet: as mentioned in the paper, we follow previous works (Uni3D, CLIP^2) for the Scannet Evaluation. Instance classification is performed on extracted object instances, not entire scenes. Objects are extracted with the instance masks provided by the dataset, and the ones with fewer than 5 points are removed. The task is zero-shot recognition on the common 17-class split (VoteNet, H3dnet). Classification logits are computed by comparing an instance's 3D features to text prompt features (e.g., "point cloud of {CLASS}").
> - ScanObjNN: As per Scannet, we adopt the benchmark from Uni3D for direct comparison, using the OBJ_ONLY variant without colors.
>
> We trust that our detailed overhead analysis, new quantitative evidence on representation quality via cosine margins, and clarifications on our experimental protocols directly address the reviewer's initial concerns and questions. We believe these additions substantially strengthen our contribution and we would be grateful if the reviewer would consider them in their final assessment of the paper. We would like to thank the reviewer again for the opportunity to elaborate on these aspects.

---

> > ### Comment · Reviewer_1m9F · 2025-08-03
> >
> > Thank you for your thorough rebuttal, which addressed most of my concerns. I have one minor suggestion: please carefully revise the next version to reduce the use of dashes (in the whole paper), and in line 169, consider changing “- and” to “, and”.

---

> > > ### Author Response · Authors · 2025-08-04
> > >
> > > We noted your comment that we addressed "most" of your concerns, and we would like to resolve any and all remaining points. If you have any additional questions or suggestions, we would be grateful for the chance to discuss them during the remaining days of the discussion period. Our goal is to ensure the final manuscript has your full support and confidence.
> > >
> > > Thank you again for your valuable engagement.

---

> > > > ### Comment · Reviewer_1m9F · 2025-08-04
> > > >
> > > > Thank you. Based on the current content of the paper, most of my questions have been addressed. However, I believe that the current focus is on some simple tasks, primarily zero-shot tasks. I suggest exploring whether the framework's text alignment improves supervised fine-tuning performance on tasks beyond zero-shot, such as classification, potentially including indoor scene tasks. While larger models may overfit, parameter-efficient fine-tuning methods [1-2] (or a discussion of PointBERT) could be beneficial.
> > > >
> > > > Incorporating these points could improve the article's visibility. I am willing to raise my score if positive results are observed. If supplementary experiments are not feasible within the rebuttal period, I will still keep a positive score and recommend including the suggested discussions in the final version.
> > > >
> > > > [1] Dynamic Adapter Meets Prompt Tuning: Parameter-Efficient Transfer Learning for Point Cloud Analysis. CVPR 24
> > > > [2] Parameter-Efficient Fine-Tuning in Spectral Domain for Point Cloud Learning. TPAMI 25

---

> ### Author Response · Authors · 2025-08-04
>
> Dear Reviewer,
>
> Thank you for your positive feedback. We are very glad to know that our rebuttal addressed most of your concerns. We will absolutely follow your suggestion and will carefully revise the use of dashes throughout the paper, including the specific change in line 169, for the final version.

---

> ### Author Response · Authors · 2025-08-06
>
> We sincerely thank you for your valuable feedback and appreciate your patience while we conducted the suggested evaluations.
>
> We agree that analyzing more supervised fine-tuning tasks is a crucial addition. We were indeed very interested in this direction and had already conducted full fine-tuning in our post-submission experiments, which we include to serve as a reference baseline. However, your suggestion to employ Parameter-Efficient Fine-Tuning methods was particularly insightful, especially for mitigating potential overfitting on large-scale models like Uni3D, which is the best but biggest transformer backbone in our work.
>
> Following your recommendation, we have conducted new experiments using multiple PEFT approaches: a traditional Adapter [1], DAPT [2], and PointGST [3]. We performed these evaluations on three key classification benchmarks: the synthetic ModelNet40, the more challenging, real-world ScanObjectNN (OBJ_ONLY as in the paper) and ScanNet object instances directly extracted from scenes through semantic masks (following the ScanNet evaluation of previous models such as Uni3D and $CLIP^2$), thus addressing also indoor scene data as requested.
>
> We based on official implementations and standard hyperparameters to ensure a fair comparison. Specifically, for the traditional Adapter, we used a bottleneck dimension of m=32 to maintain a comparable parameter count to other methods. For DAPT, we used the recommended adapter dimension of r=64. For PointGST, we used the standard adapter dimension of r=36, s=1 and k=4 local groups.
>
> Below, we present the table with the results obtained:
>
> |Model|Method|Trainable Params|ModelNet40|ScanObjectNN| ScanNet Instances |
> |:---|:---|:---|:---:|:---:|:---:|
> |Uni3D | full fine tuning | 1016.5M (100%) | 94.28 | 97.12 | 82.72 |
> |SF-Uni3D | full fine tuning | 1016.5M (100%) | 94.42 | 97.58 | 83.58 |
> |||||||
> |Uni3D | Adapter | 7.6M (0.74%) | 94.35 | 96.80 | 81.42 |
> |SF-Uni3D | Adapter | 7.6M (0.74%) | 94.46 | 97.09 | 82.56 |
> || | | | | |
> | Uni3D | DAPT | 7.3M (0.72%) | 94.33 | 96.78 | 82.65 |
> | SF-Uni3D | DAPT | 7.3M (0.72%) | 94.49 | 97.15 | 83.46 |
> | | | | | | |
> | Uni3D | PointGST | 4.11M (0.4%) | 94.83 | 97.68 | 83.04 |
> | SF-Uni3D | PointGST | 4.11M (0.4%) | 94.95 | 98.09 | 84.29 |
>
> We extract three main observations:
>
> 1. SCENEFORGE provides a stronger and more generalizable initialization.
> As clearly shown by the results, our compositional pre-training with SCENEFORGE provides a significantly more robust initialization for downstream tasks. For every fine-tuning method used, the SF-Uni3D backbone consistently outperforms the baseline Uni3D. This confirms that the richer, more diverse representations learned during pre-training are not limited to improving zero-shot capabilities but provide a solid foundation that translates into superior performance even in supervised fine-tuning scenarios. This observation is consistent with the findings already present in our paper for 3D VQA and part segmentation, where a better pretrained model led to better fine-tuned results.
>
> 2. The benefit of SCENEFORGE scales with task complexity.
> The impact of our method is modulated by the complexity of the dataset. On ModelNet40, a dataset of synthetic and "clean" objects, the performance is already near saturation, so the gain provided by SCENEFORGE is limited but still present. On ScanObjectNN, which introduces real-world noise and occlusions, the advantage of SF-Uni3D becomes more apparent. Crucially, on ScanNet Instances, involving objects extracted from complex indoor scenes, the benefit of SF-Uni3D is most pronounced. This further supports the idea that our framework is most valuable for learning robust features that can be transferred to complex real-world data.
>
> 3. Results highlight a powerful synergy between SCENEFORGE-pretraining and state-of-the-art 3D PEFT strategies. Indeed,  our pretraining allows not only full fine tuning, but also PEFT methods to reach higher performance peaks. Notably, fine tuning with PointGST obtains the best results, not only outperforming other PEFT methods but even surpassing full fine-tuning across all datasets, while using a mere 0.4% of the trainable parameters. This result validates our pretraining effectiveness and the reviewer's hint that fully fine-tuning these large models can be suboptimal, likely due to overfitting, showing that PEFT methods can be a more effective strategy for downstream tasks.
>
> We hope these experiments addressed your comments and substantially elevated the paper's contribution. We will integrate this analysis and the results into the next version.
>
> Thank you once again for your constructive participation. We hope these positive results earn your full support.
>
> [1] Parameter-Efficient Transfer Learning for NLP, ICML 19 [2] Dynamic Adapter Meets Prompt Tuning: Parameter-Efficient Transfer Learning for Point Cloud Analysis. CVPR 24 [3] Parameter-Efficient Fine-Tuning in Spectral Domain for Point Cloud Learning. TPAMI 25

---

> > ### Comment · Reviewer_1m9F · 2025-08-07
> >
> > Thanks for the response. I am going to raise the score. I hope the authors include the discussion in the final version. Good luck.

---

> > > ### Author Response · Authors · 2025-08-07
> > >
> > > Thank you so much for your positive feedback and for raising your score. We are very grateful for your support.
> > >
> > > We will absolutely ensure that the discussion and the new experimental results are integrated into the final version of the paper.
> > >
> > > Thank you again for your engagement throughout this process.

---

### Author Response · Authors · 2025-08-09
**Final Summary of Rebuttal/Discussion**

As the discussion period comes to a close, we would like to express our sincere gratitude for your reviews and discussions. We were encouraged that reviewers found our method to be effective, general and model-agnostic, with comprehensive experiments showing clear and consistent improvements on multiple downstream tasks.

We are especially grateful that this process has directly led to substantial additions that we believe significantly strengthen our paper's impact and analysis. Thanks to your insightful suggestions, we conducted and analyzed new experiments which we will integrate into the final manuscript and supplementary material. These new contributions include:

- A preliminary analysis of the point cloud sampling budget, showing how performance is affected by scene complexity and providing guidance for future work.

- A quantitative analysis focusing on the ScanQA benchmark, demonstrating our method's strong generalization to more complex, unseen spatial relations not used during pre-training.

- A comprehensive new evaluation of supervised fine-tuning on multiple classification benchmarks. These results show that our pre-training consistently boosts performance for both full fine-tuning and, crucially, several state-of-the-art Parameter-Efficient Fine-Tuning (PEFT) methods on both synthetic and challenging real-world data.

We are very pleased that the rebuttals and discussions have allowed us to thoroughly address the questions and concerns raised during the review process.

Although the discussion period is ending soon, we are still available to answer any final questions you may have.

Thank you again for your engagement. We are confident that these new results and clarifications have strengthened our work, and we hope to have earned your support.

Sincerely,

The Authors

---

### Decision · Program_Chairs · 2025-09-17

**Decision:**

Accept (poster)

**Comment:**

This paper proposes SceneForge, a framework that improves 3D-text contrastive alignment. This approach increases data diversity and improves performance on various tasks. It also enhances spatial reasoning and adapts well to more complex 3D scenes.

Strengths of this work: SceneForge is a general framework that can be easily integrated with other 3D encoders. It can improve performance on a wide range of tasks and benchmarks. The method is simple yet effective, supported by comprehensive experiments.

The rebuttal effectively addressed reviewers' major concerns and clearly explained the technical contributions of the work. As a result, the final decision is to accept the paper.